# Machine Learning Algorithm Predicts Mortality Risk in Intensive Care Unit for Patients with Traumatic Brain Injury

**DOI:** 10.3390/diagnostics13183016

**Published:** 2023-09-21

**Authors:** Kuan-Chi Tu, Eric nyam tee Tau, Nai-Ching Chen, Ming-Chuan Chang, Tzu-Chieh Yu, Che-Chuan Wang, Chung-Feng Liu, Ching-Lung Kuo

**Affiliations:** 1Department of Neurosurgery, Chi Mei Medical Center, Tainan 710402, Taiwan; gary12223@hotmail.com (K.-C.T.); wangchechuan@gmail.com (C.-C.W.); 2Department of Nursing, Chi Mei Medical Center, Tainan 710402, Taiwan; patty11@gmail.com (N.-C.C.); h85068132@yahoo.com.tw (M.-C.C.); fish5777@yahoo.com.tw (T.-C.Y.); 3Center for General Education, Southern Taiwan University of Science and Technology, Tainan 710402, Taiwan; 4Department of Medical Research, Chi Mei Medical Center, Tainan 710402, Taiwan; chungfengliu@gmail.com; 5School of Medicine, College of Medicine, National Sun Yat-sen University, Kaohsiung 804, Taiwan

**Keywords:** artificial intelligence, machine learning, traumatic brain injury, mortality, intensive care unit, computer-assisted system

## Abstract

Background: Numerous mortality prediction tools are currently available to assist patients with moderate to severe traumatic brain injury (TBI). However, an algorithm that utilizes various machine learning methods and employs diverse combinations of features to identify the most suitable predicting outcomes of brain injury patients in the intensive care unit (ICU) has not yet been well-established. Method: Between January 2016 and December 2021, we retrospectively collected data from the electronic medical records of Chi Mei Medical Center, comprising 2260 TBI patients admitted to the ICU. A total of 42 features were incorporated into the analysis using four different machine learning models, which were then segmented into various feature combinations. The predictive performance was assessed using the area under the curve (AUC) of the receiver operating characteristic (ROC) curve and validated using the Delong test. Result: The AUC for each model under different feature combinations ranged from 0.877 (logistic regression with 14 features) to 0.921 (random forest with 22 features). The Delong test indicated that the predictive performance of the machine learning models is better than that of traditional tools such as APACHE II and SOFA scores. Conclusion: Our machine learning training demonstrated that the predictive accuracy of the LightGBM is better than that of APACHE II and SOFA scores. These features are readily available on the first day of patient admission to the ICU. By integrating this model into the clinical platform, we can offer clinicians an immediate prognosis for the patient, thereby establishing a bridge for educating and communicating with family members.

## 1. Introduction

Traumatic brain injury is a global issue that not only impacts patients’ health but also imposes a significant burden on social, economic, and medical resources [1]. The age-adjusted mortality rate in Europe is 11.7 per 100,000 and 17.0 per 100,000 in the US [2,3]. In contrast to Western countries, where TBI is often associated with war, Asia experiences TBI due to falls and road traffic injuries [4]. As low- to middle-income countries undergo industrial transformation leading to increased mechanization and urbanization, the incidence of brain injuries is gradually rising. However, the slow growth of medical resources in these countries results in more severe disabilities compared to developed nations [4].

Survivors of TBI typically face neurological deficits and disabilities. Those with severe TBI receive treatment in the intensive care unit (ICU). Various efforts have been made to predict the prognosis of TBI patients, exploring factors such as Glasgow Coma Scale (GCS), age, pupillary reactivity, injury severity, and clinical condition (e.g., hypoxia, respiratory distress, and hypotension) in numerous studies. The evaluation of brain injury extent and classification using CT scans is also closely linked to mortality [5,6,7,8].

A previous retrospective study found variability in the use of a single predictive model across populations [9]. Although studies of IMPACT and CRASH are widely known, they may not be applicable to each individual patient [10]. The SOFA (Sequential Organ Failure Assessment), introduced in 1996, is designed to describe the progression of complications in critically ill patients and an elevated SOFA score is associated with a higher likelihood of mortality [11,12]. APACHE II relies on 12 physiological variables measured within the first 24 h of ICU admission to predict ICU patient outcomes [13]. However, the use of APACHE II and SOFA has only shown marginal improvement in prognostic performance [14]. Therefore, we need to seek more accurate predictive models for prognosis and mortality in ICU settings.

Machine learning (ML) approaches require more input and output data for analysis, but they excel at handling complex interrelationships. Compared to classical linear regression statistics, machine learning processes data directly, resulting in more accurate predictions [15]. However, the “black-box” nature of AI, characterized by its lack of explanation, is still the main reason for the low clinical application. In order to improve the predictive explanation of AI models, Explanatory Artificial Intelligence (XAI) techniques have been introduced, with SHAP (SHapley Additive exPlanations) being the most widely used XAI technique for explaining which clinical features are important for predicting various diseases or patient prognosis. Therefore, it is very important to use XAI to better interpret how each feature contributes to the associated outcome in the AI prediction model [16].

Courville E et al. reported a systematic view and meta-analysis (2013–2020) demonstrating that much of this literature discusses in-hospital mortality and poor prognosis, but lacks a more specific focus on the ICU population to understand the predictive power of AIs in TBI patients [17]. In the last three years, there have been several reports on the prognosis and mortality risk of brain injury using ML techniques. However, some of these studies may not have selected different combinations of features based on clinical importance, lacked comparisons with traditional tools, or were not conducted in an ICU setting. Therefore, further investigation is needed to clarify this point [18,19,20,21].

Our goal is to use machine learning algorithms to analyze the vast amount of ICU data to predict mortality risk after TBI, which is more tailored to patients in our country. Additionally, it is essential to compare these ML models with the existing APACHE II and SOFA scores. We also use the SHAP technique to explain which clinical features are important for predicting various diseases or patient outcomes.

## 2. Materials and Methods

### 2.1. Ethics

This research received ethical approval (revision: 11106-013) from the institutional review board at Chi Mei Medical Center in Tainan, Taiwan. The authors conducted the study in accordance with appropriate guidelines and regulations. Since the study was retrospective in nature, the Ethics Committee waived the requirement for informed consent.

### 2.2. Flow Chart and AI Device of Current Study

Our study followed the guidelines specified in the Transparent Reporting of a Multivariable Prediction Model for Individual Prognosis or Diagnosis (TRIPOD) standard. Figure 1 illustrates the flowchart detailing the ML training process and its integration into the hospital system. The ML model was trained using a total of 42 selected features identified based on their statistically significant differences (*p*-value < 0.05) between the mortality and non-mortality groups.

To assess model performance, a 70% training dataset was used, while the remaining 30% formed the test set via random splitting. As a result, four models were developed to predict mortality risk.

Statistical analysis involved *t*-tests for numerical variables and Chi-square tests for categorical variables. Additionally, Spearman correlation analysis was conducted to evaluate the strength of the correlation between each feature and the outcome. Recognizing the imbalanced outcome classes, particularly in mortality cases, we employed the synthetic minority oversampling technique (SMOTE) [22]. This oversampling technique was applied to balance the number of positive outcome cases (mortality) with the negative cases (survival) during the final model training with each machine learning algorithm.

Figure 2 illustrates the utilization of the hospital backend system to collect data from various assessment modules, including the ICU evaluation module, vital signs module, health status module, and medical history module. These modules provide input to the central computer for integrated processing, and the data are then fed into the ML training model for simulation.

### 2.3. Patient Selection

From January 2016 to December 2021, a retrospective collection of patients aged 20 years and older who were diagnosed with TBI and admitted to the ICU was conducted using the electronic medical records of Chi Mei Medical Center. The inclusion criteria included neurosurgical patients who have been admitted to the ICU with the following diagnostic codes. ICD-9: 800*–804*, 850*–854*, 959.0, 959.01, and 959.8–959.9; ICD-10: S00*-T07*. Patients with missing or ambiguous values were excluded.

### 2.4. Feature Selection and Model Building

Under the consensus of several neurosurgeons and intensive care physicians, we identified parameters that met the following criteria: (1) representation of the clinical status of traumatic brain injury patients, (2) objective assessability, and (3) generalizability. Subsequently, we employed univariate filter methods for feature selection, considering both continuous and categorical variables. A significance level of 0.05 or lower was used for selection. Additionally, Spearman’s correlation coefficient and expert opinions were considered during the finalization of the feature selection process. The study utilized 42 features, as listed in Table 1. We employed four machine learning algorithms, including Logistic Regression [23], Random Forest [24], LightGBM [25], and XGBoost [26], to construct predictive models for mortality in ICU. To reduce concerns of overfitting that might arise from a small dataset, we utilized the cross-validation technique to build the models.

### 2.5. Model Performance Measurement

In this study, we evaluated the performance of the machine learning models using accuracy, sensitivity, specificity, and area under the curve (AUC) of the receiver operating characteristic curve (ROC).

Specificity is an important metric to assess the ability of a test or diagnostic method to correctly identify normal results (non-patients), while sensitivity evaluates the ability to correctly identify positive outcomes (patients). These metrics are mutually influencing and should be considered comprehensively in research [27].

Accuracy measures the correctness of predictions made by a classification model or testing method and represents the proportion of correct predictions among all predictions made. However, in certain imbalanced datasets, accuracy can be misleading and lead to poor prediction performance for minority classes [28].

The AUC, representing the area under the ROC curve, which represents the trade-off between sensitivity and specificity (false positive rate) at different thresholds, serves as an effective “summary” of the ROC curves’ performance [29,30].

To assess the superiority of each machine learning model compared to traditional tools, we specifically used the DeLong test [31].

## 3. Results

### 3.1. Characteristics and Clinical Presentations of Individuals with Traumatic Brain Injury

A total of 2260 patients were retrospectively included from the electronic medical records system of Chi-Mei Hospital. Among them, there were 1447 males (64.03%) and 813 females (35.97%). The average age was approximately 63.89 ± 17.74 (mean ± SD) years old. The characteristics of the patients are listed in Table 1, comprising 42 features, including vital signs, coma scale, pupillary reflex, intubation status, external ventricular drainage, and comorbidities. Among these, 29 features showed a significant difference in relation to mortality (*p*-value < 0.05).

### 3.2. The Correlation between Factors and Mortality (Spearman Correlation Coefficient)

To accurately quantify the impact of each factor on prediction within the ML model, we conducted an analysis using the Spearman correlation coefficient. Among the factors, 22 had coefficients greater than 0.1 (italic) and showed a significant correlation with mortality, indicating their substantial influence on prediction. Moreover, among these features, 14 had coefficients greater than 0.2 (bold) and demonstrated a significant correlation with mortality (Table 2). The top five variables exhibiting high correlation coefficients include pupil_reflex + (R), pupil_reflex + (L), vasopressors, GCS_M, and GCS_E. Notably, while SOFA and APACHE II were employed to compare predictive performances with the AI model, they were not utilized as features in the AI model itself.

### 3.3. Predictive Models with Different Features Combinations

Table 3 presents the predictive outcomes obtained from various feature combinations and artificial intelligence learning. Initially, there were 42 features, which were then categorized based on their significant difference with mortality and their Spearman correlation coefficient. This resulted in three groups: 29 features significantly correlated with mortality, 22 features with a Spearman correlation coefficient greater than 0.1, and 14 features with a Spearman correlation coefficient greater than 0.2. It should be noted that the original 15-feature model includes four features: GCS_E, GCS_V, GCS_M, and GCS. Since GCS is the sum of GCS_E, GCS_V, and GCS_M, we therefore excluded the GCS feature and built the 14-feature model. The results show that the impact on the model’s quality is not significant.

Each feature combination was assessed across four different machine learning models, and the performance of each model was evaluated using the AUC of the ROC curve to determine the best predictive model. Regardless of the feature combination, the best-performing machine learning model achieved an AUC greater than 0.9.

Among the 42 features, the LightGBM model performed the best with an AUC of 0.916. In the combination of 29 features, the Random Forest model achieved the highest AUC of 0.918. For the 22-feature combination, the Random Forest model again outperformed others with an AUC of 0.921. Lastly, in the combination of 14 features, the LightGBM model had the highest AUC of 0.914 (Figure 3a–d).

### 3.4. Comparing the Best-Performing Model with Traditional ICU Assessment Tools in Different Feature Combinations

In the DeLong test, no significant differences (>0.05) were observed in any of the feature combinations when compared to the combination of 42 features and the LightGBM model. For the sake of clinical convenience, we believe that using a combination of 14 features is easier to execute. When compared to APACHE II and SOFA scores, the *p*-values obtained were 0.0180 and 0.0156, respectively, indicating significant differences (Table 4).

### 3.5. Feature Importance of AI Algorithm LightGBM Using 14 Feature Variables

Feature importance was used to rank the most important attributes that significantly contribute to the accuracy of the final prediction models [32]. To better interpret how each feature contributes to the associated outcome, we performed SHAP (SHapley Additive exPlanations) [33].

We ranked the significance of all variables in the LightGBM model to comprehend the role of each better (Figure 4). In Figure 4a, the color of the SHAP plot represents the size of the original feature values, with red indicating positive variable values and blue indicating negative ones. The SHAP value signifies the degree of a feature’s impact on the outcome (a positive SHAP value indicates a positive effect). A wider Feature SHAP value suggests a more extensive influence on the outcome. As depicted, patients using vasopressors (represented by red dots) have an increased risk of death (SHAP value is positive), whereas the impact of GCS_M and GCS_V is the opposite. Figure 4b displays the ranking of features’ influence on the outcome based on the absolute values of the SHAP values. The figure shows that the top five influential feature variables are vasopressors, GCS_M, GCS_V, pupil reflex + (R), and Muscle_RLE.

Based on the contribution of each predictor to the machine learning method, it can be presented in the form of feature importance (Figure 4).

### 3.6. Integration and Application of AI with Clinical Systems

After a series of analyses, we concluded that the LightGBM model with a combination of 14 features was more lightweight. Therefore, we integrated it into the hospital system to assist clinical doctors and nurses in treatment and facilitate communication with patients’ families. The “Original” column represents data for current status. Currently, it displays data from the time of admission to the ICU. The “Adjust” column allows the observer to adjust the values of each feature to understand the effect of each feature on the risk of mortality as a reference for treatment. (Figure 5).

## 4. Discussion

This is the first study to demonstrate the mortality risk of TBI in ICU using a machine learning model and compare it to the present prediction model. The novelty of the current study is as follows. The simplified model using 14 features with the LightGBM algorithm for mortality prediction proved to be the most practical and excellent, achieving an AUC of 0.914. The study made significant achievements in several aspects: (a) specialized ICU parameters improved the credibility of prediction results; (b) different feature combinations were chosen based on clinical importance and correlation with mortality significance; (c) a comparison was made between ML techniques and commonly used ICU prognostic indicators and mortality assessment tools, such as APACHE II and SOFA scores (4). The observer can adjust the values of each feature to understand the effect of each feature on the risk of mortality as a reference for treatment.

This study employed artificial intelligence (AI) for data analysis, offering numerous advantages. ML can handle complex interactions in vast datasets, leading to more accurate outcome predictions. However, ML models require a larger number of input-output pairings for training, and interpretability may be sacrificed compared to standard statistics [18]. In this study, we utilized AI to identify suitable models and clinically examine the mortality of patients with brain injury admitted to the ICU.

The data from 2260 patients, including electronic medical records, clinical physiological values, and laboratory tests, were collected and analyzed. Initially, 42 features were included, but not all of them showed a correlation with mortality. Therefore, we performed a direct analysis of the features and mortality, comparing their significance, and found that 29 parameters exhibited a significant difference in relation to mortality as Table 1 shows. Further analysis involved considering Spearman’s correlation coefficient values, which led us to identify 14 features from LightGBM that still possessed a high AUC, making it the most accurate prediction model. Utilizing the mortality risk provided by AI, clinicians can be assisted in making informed medical decisions.

At our hospital, we primarily use the APACHE II and SOFA assessment tools to assist with clinical decision-making and effectively communicate with patients and their families to explain their medical condition in the ICU. Despite the existence of more precise and updated versions such as APACHE III and IV, APACHE II continues to be the predominant severity grading system and mortality risk in use [34]. The SOFA score is also widely used by critical-care physicians due to its ability to provide rapid and accurate mortality predictions [35]. To compare the AI models with APACHE II and SOFA scores, we employed the DeLong test. The results revealed that the ML models generally outperformed the traditional tools. This finding suggests the potential clinical utility of AI in this study. For ease of clinical practice and completeness of data acquisition, we chose to use a 14-feature LightGBM predictive model for clinical use.

Figure 4 shows that the use of vasopressors predominated and significantly influenced the mortality risk in the LightGBM model. Maintaining the stability of mean arterial pressure and cerebral perfusion pressure (CPP) has always been crucial in brain injury care. The judicious use of vasopressors helps balance intracranial pressure and maintain a constant CPP [36]. For intubated patients, motor evaluation was relatively more important due to the inability to assess verbal function. The focus was primarily on the unaffected side’s functionality to determine the patient’s prognosis [37]. A GCS score below 8 indicates severe brain injury, often requiring intubation to protect the airway. According to the study by Hsu SD et al., not only GCS but also systolic blood pressure (SBP) is an important prognostic factor. In the emergency department, if a patient has a GCS < 6 or an SBP < 84 mmHg, immediate life-saving measures need to be taken [19,38]. Monitoring blood pressure and tracking changes in the GCS can be beneficial for predicting prognosis. However, in Hsu SD’s study [19], they utilized features from the emergency department, whereas we utilized features from the ICU, where patients have already received treatment. Consequently, the mortality risk prediction based on ICU features tends to be more accurate at that stage.

Table 5 presents the literature comparison we conducted. In comparison to other literature, our study examines the impact of different feature combinations on mortality risk prediction and suggests that the predictive capability of the machine learning model outperforms traditional tools (APACH II, and SOFA scores). In addition, the model is currently being applied in ICU. We believe that this model can serve as an alternative choice for routine assessment in the ICU.

Generally, IMPACT and CRASH are commonly used prognostic tools for predicting outcomes and mortality in clinical TBI cases [39,40]. In Han J et al.’s report, these two traditional tools were found to have an AUC of 0.86 and 0.87, which is significantly lower compared to our ML approach [41]. Wu X et al. compared XGBoost, a machine learning algorithm, with traditional prediction tools such as IMPACT and CRASH. The results demonstrated that machine learning (ML), specifically XGBoost, outperformed IMPACT and CRASH the traditional tools in terms of predictive accuracy [21]. In Table 5, our AUC is greater than Wu’s model, indicating that our model is more suitable for clinical use.

Moreover, the AI predictive tool we propose is intended as a clinical aid, not a replacement for a doctor’s judgment. Before implementing policies based on AI predictions, it is essential to conduct comprehensive evaluations in terms of ethics, society, and policy. For example, protecting patients’ data privacy and rights and ensuring they are not treated unfairly because of AI predictions.

Despite the robust ML algorithms demonstrating promising predictive performance, this study still has some limitations. First, it is a retrospective study, and prospective research is needed to validate the experimental results. Second, the diagnosis of brain injuries relies on Taiwan’s National Health Insurance regulations, which may have a small number of miscoded diagnosis codes. However, the impact of these miscodings is relatively minor in terms of overall influence. Third, imaging parameters such as midline shift and presence/absence of brain ventricles have not been quantitatively incorporated into our ML model. Fourth, the potential confounding effects of the numerous features utilized require further exploration. Fifth, additional confounding variables such as smoking, alcohol intake, shifts in treatment guidelines, and emerging medical practices could not be comprehensively assessed due to the constraints of the retrospective database. Last, the current ML training is limited to various medical centers and laboratories, and due to differences in treatment guidelines, the generalization of ML from a single center to other regions is not yet possible. However, we provide the logical framework for ML, and the iterative process validates the effectiveness and value of such predictive models. Based on this foundation, further research can be conducted to improve upon these findings.

## 5. Conclusions

Our research primarily focuses on training AI using ICU data and utilizing various feature combinations to identify suitable ML models. In the end, we obtained 14 feature combinations (with a significant correlation to mortality and Spearman > 0.2), among which LightGBM performed exceptionally well. Not only does it demonstrate mortality prediction capabilities on par with models using more features but it also outperforms traditional models. These research findings can be applied in critical clinical settings to assist physicians in assessing patients’ conditions and providing more data-driven explanations during communication with family members. In the future, we advocate for more studies that focus on incorporating additional variables to enhance model performance. The application of AI predictions in other healthcare settings, such as emergency care and long-term care, warrants deeper exploration.

## Figures and Tables

**Figure 1 diagnostics-13-03016-f001:**
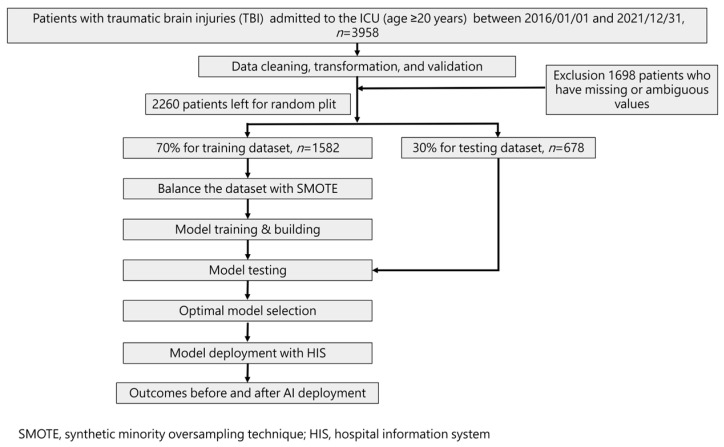
Workflow diagram for data collection and machine learning model training.

**Figure 2 diagnostics-13-03016-f002:**
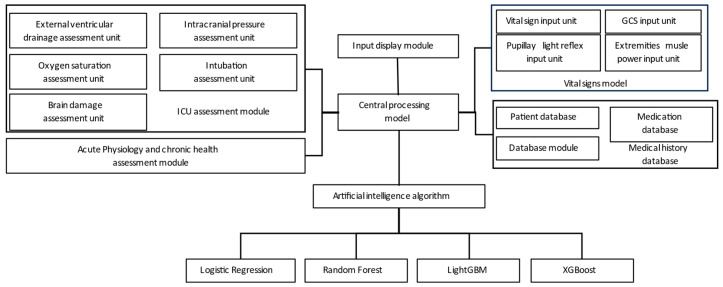
System training architecture.

**Figure 3 diagnostics-13-03016-f003:**
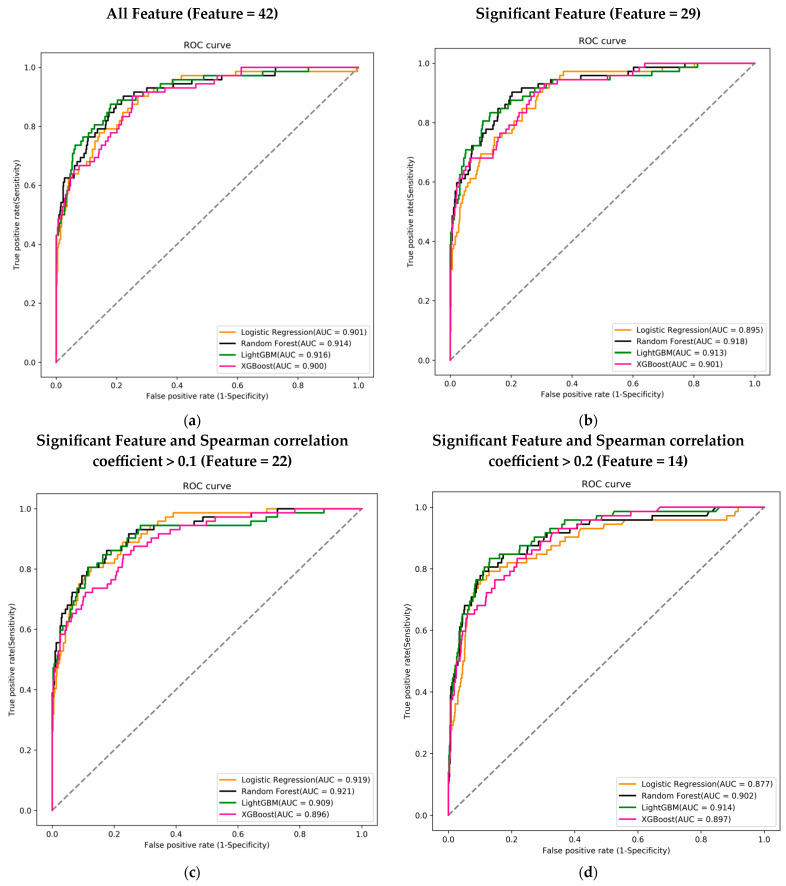
Receiver operating characteristic curves (ROC), area under the curve (AUC), for mortality prediction in the training course. (**a**) Using 42 features to train the ML model; (**b**) using 29 features that were significant in the mortality; (**c**) using 22 features that were significant and Spearman correlation coefficient >0.1; and (**d**) using 14 features that were significant and Spearman correlation coefficient >0.2. Logistic regression (LR) (orange), random forest (black), LightGBM (green), and XGBoost (pink) using the 14 feature variables.

**Figure 4 diagnostics-13-03016-f004:**
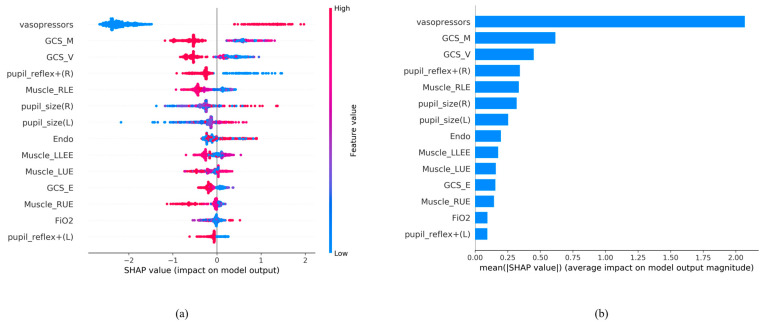
SHAP analysis results. (**a**) SHAP global explanation on the 14-feature model (LightGBM model); (**b**) SHAP absolute value of each feature on the 14-feature model (LightGBM model).

**Figure 5 diagnostics-13-03016-f005:**
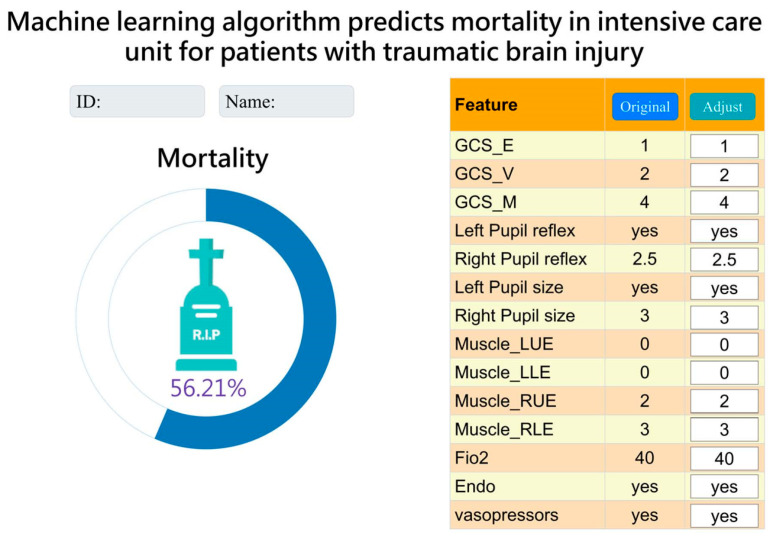
Interface presentation of AI in practical application within the Chi Mei Hospital healthcare system.

**Table 1 diagnostics-13-03016-t001:** Characteristics and significance of traumatic brain injury patients.

Feature	Overall *n* = 2260	Non-Mortality *n* = 2020	Mortality *n* = 240	*p*-Value
Female, *n* (%)	813 (35.97)	735 (36.39)	78 (32.50)	0.265
male, *n* (%)	1447 (64.03)	1285 (63.61)	162 (67.50)	
Age, mean (SD)	63.89 (17.74)	63.26 (17.76)	69.22 (16.65)	<0.001
height, mean (*SD*)	162.74 (11.24)	162.75 (10.95)	162.60 (13.43)	0.862
weight, mean (SD)	63.00 (14.16)	63.24 (14.23)	61.00 (13.42)	0.016
Systolic blood pressure (SBP), mean (*SD*)	142.36 (29.41)	143.33 (28.40)	134.22 (35.86)	<0.001
Diastolic blood pressure (DBP), mean (*SD*)	78.02 (17.02)	78.72 (16.36)	72.13 (20.90)	<0.001
Mean Arterial Pressure (MAP), mean (SD)	100.04 (20.67)	100.99 (19.86)	92.06 (25.24)	<0.001
Body temperature (BT), mean (*SD*)	36.55 (0.63)	36.57 (0.56)	36.39 (1.01)	0.005
pulse, mean (*SD*)	86.48 (16.95)	85.93 (15.90)	91.10 (23.57)	0.001
Respiratory rate (RR), mean (*SD*)	17.67 (4.06)	17.73 (3.95)	17.10 (4.83)	0.054
Glasgow Coma Scale_eye opening (GCS_E), mean (*SD*)	3.13 (1.26)	3.31 (1.15)	1.69 (1.18)	<0.001
Glasgow Coma Scale_verbal response (GCS_V), mean (*SD*)	3.52 (1.75)	3.75 (1.66)	1.65 (1.30)	<0.001
Glasgow Coma Scale_motor response (GCS_M), mean (*SD*)	4.99 (1.77)	5.21 (1.60)	3.08 (1.97)	<0.001
Glasgow Coma Scale (GCS), mean (*SD*)	11.64 (4.48)	12.27 (4.11)	6.41 (4.03)	<0.001
Left Pupil				
Pupil reflex (−), *n* (%)	230 (10.18)	104 (5.15)	126 (52.50)	<0.001
Pupil reflex (+), *n* (%)	2030 (89.82)	1916 (94.85)	114 (47.50)	
Pupil size (L), mean (*SD*)	3.23 (0.99)	3.10 (0.77)	4.29 (1.70)	<0.001
Right Pupil				
Pupil reflex (−), *n* (%)	231 (10.22)	103 (5.10)	128 (53.33)	<0.001
Pupil reflex (+), *n* (%)	2029 (89.78)	1917 (94.90)	112 (46.67)	
Pupil size (R), mean (*SD*)	3.22 (0.99)	3.09 (0.76)	4.34 (1.74)	<0.001
Muscle power_left upper extremity (Muscle_LUE), mean (*SD*)	3.03 (1.66)	3.24 (1.54)	1.30 (1.59)	<0.001
Muscle power_left lower extremity (Muscle_LLEE), mean (*SD*)	2.93 (1.67)	3.13 (1.58)	1.24 (1.48)	<0.001
Muscle power_right upper extremity (Muscle_RUE), mean (*SD*)	3.04 (1.66)	3.25 (1.54)	1.30 (1.57)	<0.001
Muscle power_right lower extremity (Muscle_RLE), mean (*SD*)	2.94 (1.67)	3.14 (1.58)	1.22 (1.46)	<0.001
Inspired fraction of oxygen (FiO2), mean (*SD*)	27.80 (11.52)	26.49 (9.08)	38.84 (20.50)	<0.001
APACHE II, mean (*SD*)	12.92 (7.44)	11.71 (6.44)	23.10 (7.49)	<0.001
Sequential Organ Failure Assessment (SOFA score), mean (*SD*)	3.10 (2.72)	2.64 (2.26)	6.94 (3.17)	<0.001
Endotracheal tube (Endo)				
No, *n* (%)	1283 (56.77)	1229 (60.84)	54 (22.50)	<0.001
Yes, *n* (%)	977 (43.23)	791 (39.16)	186 (77.50)	
External ventricular drain (EVD)				
No, *n* (%)	2045 (90.49)	1823 (90.25)	222 (92.50)	0.313
Yes, *n* (%)	215 (9.51)	197 (9.75)	18 (7.50)	
Intracranial pressure (ICP), *n* (%)				
No, *n* (%)	2025 (89.60)	1835 (90.84)	190 (79.17)	<0.001
Yes, *n* (%)	235 (10.40)	185 (9.16)	50 (20.83)	
Cerebral perfusion pressure (CPP), *n* (%)				
No, *n* (%)	2025 (89.60)	1835 (90.84)	190 (79.17)	<0.001
Yes, *n* (%)	235 (10.40)	185 (9.16)	50 (20.83)	
surgery, *n* (%)	310 (13.72)	247 (12.23)	63 (26.25)	<0.001
Drugs				
vasopressors, *n* (%)	293 (12.96)	157 (7.77)	136 (56.67)	<0.001
sedative_hypnotic, *n* (%)	950 (42.04)	787 (38.96)	163 (67.92)	<0.001
Perdipine, *n* (%)	354 (15.66)	295 (14.60)	59 (24.58)	<0.001
Underlying disease				
Hypertension, *n* (%)	954 (42.21)	829 (41.04)	125 (52.08)	0.001
Diabetes mellitus, *n* (%)	581 (25.71)	510 (25.25)	71 (29.58)	0.169
heart disease, *n* (%)	363 (16.06)	320 (15.84)	43 (17.92)	0.462
Cerebrovascular disease, *n* (%)	206 (9.12)	181 (8.96)	25 (10.42)	0.534
Gastrointestinal disease, *n* (%)	168 (7.43)	151 (7.48)	17 (7.08)	0.929
Liver Disease, *n* (%)	161 (7.12)	135 (6.68)	26 (10.83)	0.026
kidney disease, *n* (%)	133 (5.88)	100 (4.95)	33 (13.75)	<0.001
cancer, *n* (%)	110 (4.87)	97 (4.80)	13 (5.42)	0.795
Thyroid disease, *n* (%)	55 (2.43)	53 (2.62)	2 (0.83)	0.139
epilepsy, *n* (%)	45 (1.99)	40 (1.98)	5 (2.08)	0.809
asthma, *n* (%)	41 (1.81)	39 (1.93)	2 (0.83)	0.310
pneumonia, *n* (%)	38 (1.68)	32 (1.58)	6 (2.50)	0.286

Note. A *t*-test was used for numerical variables and the Chi-square test was used for categorical variables. Surgical procedures are as follows: decompressive craniectomy, acute epidural hematoma removal, acute subdural hematoma removal, acute intracerebral hematoma removal, and intracranial pressure monitor placement. A patient who undergoes one of the above five surgical procedures is said to have undergone surgery.

**Table 2 diagnostics-13-03016-t002:** The Spearman correlation coefficient for each factor.

Feature	Mortality	Feature	Mortality
Gender	0.025	**FiO2**	**0.294**
*Age*	*0.108*	**APACHE II**	**0.397**
Hight	0.008	**SOFA**	**0.398**
Weight	−0.045	**Endo**	**0.238**
SBP	−0.066	EVD	−0.024
*DBP*	*−0.108*	*ICP*	*0.118*
*MAP*	*−0.110*	*CPP*	*0.118*
BT	−0.088	*surgery*	*0.126*
pulse	0.079	**vasopressors**	**0.448**
RR	−0.066	*Sedative−hypnotic drugs*	*0.181*
**GCS_E**	**−0.371**	Perdipine	0.085
**GCS_V**	**−0.348**	Hypertension	0.069
**GCS_M**	**−0.398**	Diabetes mellitus	0.031
**GCS**	**−0.363**	Cerebrovascular disease	0.016
**pupil_reflex + (L)**	**−0.483**	heart disease	0.017
**pupil_size(L)**	**0.235**	asthma	−0.025
**pupil_reflex + (R)**	**−0.491**	pneumonia	0.022
**pupil_size(R)**	**0.241**	Gastrointestinal disease	−0.005
**Muscle_LUE**	**−0.325**	cancer	0.009
**Muscle_LLEE**	**−0.326**	Liver Disease	0.050
**Muscle_RUE**	**−0.328**	epilepsy	0.002
**Muscle_RLE**	**−0.331**	*kidney disease*	*0.115*
		Thyroid disease	−0.036

Note. *Italicized* text: absolute value greater than 0.1; **Bold** text: absolute value greater than 0.2.

**Table 3 diagnostics-13-03016-t003:** Model performance with different feature combinations.

Algorithm	Accuracy	Sensitivity	Specificity	AUC
42 features				
Logistic Regression	0.799	0.806	0.799	0.901
Random Forest	0.829	0.833	0.828	0.914
**LightGBM**	**0.832**	**0.833**	**0.832**	**0.916**
XGBoost	0.794	0.806	0.792	0.900
29 significant features				
Logistic Regression	0.771	0.833	0.764	0.895
**Random Forest**	**0.844**	**0.847**	**0.843**	**0.918**
LightGBM	0.835	0.833	0.835	0.913
XGBoost	0.783	0.792	0.782	0.901
22 significant features and Spearman correlation coefficient > 0.1				
Logistic Regression	0.833	0.819	0.835	0.919
**Random Forest**	**0.830**	**0.833**	**0.830**	**0.921**
LightGBM	0.851	0.819	0.855	0.909
XGBoost	0.785	0.806	0.782	0.896
14 significant features and Spearman correlation coefficient > 0.2				
Logistic Regression	0.814	0.819	0.814	0.877
Random Forest	0.832	0.833	0.832	0.902
**LightGBM**	**0.8** **78**	**0.8** **06**	**0.8** **86**	**0.91** **4**
XGBoost	0.794	0.806	0.794	0.897

Note. AUC = Area under receiver operating characteristic curve. Algorithms in bold indicate the model with the highest AUC.

**Table 4 diagnostics-13-03016-t004:** The DeLong test of ML models with different feature combinations and conventional tools (APACH II and SOFA scores).

Algorithm	Accuracy	Sensitivity	Specificity	AUC	Delong Test
Feature = 42 (LightGBM)	0.832	0.833	0.832	0.916	-
Feature = 29 (Random Forest)	0.844	0.847	0.843	0.918	0.8376
Feature = 22 (Random Forest)	0.830	0.833	0.830	0.921	0.5641
Feature = 14 (LightGBM)	0.878	0.806	0.886	0.914	0.8198
APACH II	0.768	0.847	0.759	0.872	0.0180
SOFA	0.801	0.778	0.804	0.853	0.0156

**Table 5 diagnostics-13-03016-t005:** A comparative analysis of the mortality rate among patients with brain injury over the past five years, as reviewed in our study.

Study	Current Study, 2023	Abujaber et al. [18], 2020	Hsu et al. [19], 2021	Wang et al. [20], 2022	Wu et al. [21], 2023
Setting	ICU	In-hospital	In-hospital	In-hospital	In-hospital
Patient number	2260	1620	3331	368	2804
Study models	Four ML models	Two ML models	Seven ML models	Two ML models	4 ML models
Features	Different features (42, 29, 22, 14) combination	20	8	21	26
Outcome	Mortality	Mortality	Mortality	Mortality	Mortality
Testing result (AUC)	0.915	0.96	0.82	0.955	0.87
Comparing with other prediction models	APACHE II score, SOFA score	Nil.	Nil.	Nil.	IMPACT, CRASH
The best prediction model	LightGBM (14 features)	SVM	J48	XGBoost	XGBoost

## Data Availability

Based on the privacy of patients within the Chi Mei Medical Center’s Health Information Network, the primary data underlying this article cannot be shared publicly. However, de-identified data will be shared upon reasonable request to the corresponding author.

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
