# Peer review of "Machine Learning Algorithm Predicts Mortality Risk in Intensive Care Unit for Patients with Traumatic Brain Injury"

_diagnostics, 2023, doi:10.3390/diagnostics13183016_

Round 1

Reviewer 1 Report

1. The article's unclear explanation of the inclusion and exclusion criteria raises questions about how the study's patients were chosen. Retrospective data collection from patients admitted to the ICU with traumatic brain injury is mentioned in the article. The inclusion and exclusion criteria that were used to choose the study's patients aren't, however, covered in detail. What exactly were these criteria, and why were they chosen?

2. Because the article skips over data quality control techniques, there is uncertainty regarding the accuracy and dependability of the data used. How was data quality control carried out, especially in light of the study's retrospective nature? What steps were taken to reduce data errors or selection bias?

3. The article leaves out details about the precise surgical procedures carried out on patients with traumatic brain injury, leaving a gap in our knowledge of this crucial factor. The article lists mortality rates for traumatic brain injury patients, but it is unclear if these patients underwent particular surgical procedures like decompressive surgery. What was the connection between the study's observed mortality rates and surgical interventions?

4. The statistical analysis' use of multiple comparison corrections is not mentioned in the article, leaving a hole in how the findings should be interpreted. The use of statistical tests to compare patient groups is mentioned in the article, but it is unclear whether multiple comparison corrections, like the Bonferroni method, were used. Was there any type I error control implemented?

5. It is difficult to comprehend the clinical significance of the associations discovered in the statistical analyses because the article does not discuss the associations' effect sizes. The statistical findings show that there are large differences between the groups, but they don't discuss the magnitude of the associations. What was the analysis's effect size, and what does it mean clinically?

6. There is a lack of clarity regarding potential ethical conundrums because the article doesn't discuss ethical issues related to the implementation of public health policies based on the results. Based on the findings, the article makes policy recommendations for politics and public health. It does not, however, address any potential ethical issues that might arise from the application of these policies. What ethical factors were taken into account when making recommendations based on the study's findings?

7. Because the generalizability of the findings to healthcare contexts outside of intensive care units (ICUs) is not addressed in the article, the conclusions are only partially applicable to other public health settings. The study only discusses the implications of these findings for patients receiving care in other healthcare settings, such as smaller hospitals, but it only focuses on patients who are admitted to the ICU. What are the results' generalizability to various healthcare contexts?

8. The discussion section of the article does, in fact, give scant information about the limitations of the study, particularly with regard to possible sources of bias. For a complete assessment of the study's robustness, more information on this subject is required. Limitations are briefly mentioned in the discussion section, but sources of bias or confounding variables are not fully explored. Could you provide more information about the study's restrictions, in particular regarding any possible bias sources that might have affected the findings?

I appreciate being chosen to review this article. I applaud the authors for their significant advancements in the field.

 "The data was analyze." Correction: "The data were analyzed."

  "A study focused on patients in ICU." Correction: "A study focused on the patients in the ICU."

 "The results showed a significant p-value, and a strong effect." Correction: "The results showed a significant p-value and a strong effect."

"The study was conducted in a period of time." Correction: "The study was conducted during a period."

 "The authors examined the brain injuries in the patients who were in the ICU." Correction: "The authors examined the brain injuries in ICU patients."

Author Response

Dear editor: 

Please see the attachment. Thanks.

Reviewer 2 Report

I would like to thank the authors for this interesting work. The article is well-written, and the methodology is largely clear. I only have a few points to consider, please.

(1)

In the introduction, there is a need for additional clarification on the specific motivations of the research. By providing more context and detail on these aspects, the authors can help readers to understand the relevance and the potential implications of the study's findings. For example, does literature generally lack such studies?

(2)

With the increasing adoption of Explainable AI (XAI), I find it necessary to refer to part of such contributions. The introduction can include examples of XAI applications to predict patient mortality, for example:

https://doi.org/10.1109/ICHI48887.2020.9374393

(3)

The references of XGBoost and Random Forest should be cited, please.

Chen, T., & Guestrin, C. (2016, August). Xgboost: A scalable tree boosting system. In Proceedings of the 22nd ACM SIGKDD International Conference on Knowledge Discovery and Data Mining (pp. 785-794).

Breiman, L. (2001). Random forests. Machine Learning, 45, 5-32.

(4)

Please discuss any possible limitations of the study including the (relatively) small dataset.

(5)

For the future work, I suggest exploring the use of explainable ML using popular frameworks such as SHAP or LIME. It might be interesting to compare the findings of the present work with explanations provided by those frameworks.

The quality of language is generally good.

Author Response

(The authors gave the same response as above.)
